# FLEMING-R1: TOWARD EXPERT-LEVEL MEDICAL REASONING VIA REINFORCEMENT LEARNING

## ABSTRACT

While large language models show promise in medical applications, achieving expert-level clinical reasoning remains challenging due to the need for both accurate answers and transparent reasoning processes. To address this challenge, we introduce Fleming-R1, a model designed for verifiable medical reasoning through three complementary innovations. First, our Reasoning-Oriented Data Strategy (RODS) combines curated medical QA datasets with knowledge-graph-guided synthesis to improve coverage of underrepresented diseases, drugs, and multi-hop reasoning chains. Second, we employ Chain-of-Thought (CoT) cold start to distill high-quality reasoning trajectories from teacher models, establishing robust inference priors. Third, we implement a two-stage Reinforcement Learning from Verifiable Rewards (RLVR) framework using Group Relative Policy Optimization, which consolidates core reasoning skills while targeting persistent failure modes through adaptive hard-sample mining. Across diverse medical benchmarks, Fleming-R1 delivers substantial parameter-efficient improvements: the 7B variant surpasses much larger baselines, while the 32B model achieves near-parity with GPT-4o and consistently outperforms strong open-source alternatives. These results demonstrate that structured data design, reasoning-oriented initialization, and verifiable reinforcement learning can advance clinical reasoning beyond simple accuracy optimization. We release Fleming-R1 publicly to promote transparent, reproducible, and auditable progress in medical AI, enabling safer deployment in high-stakes clinical environments.

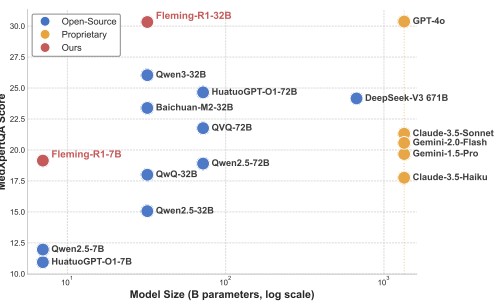
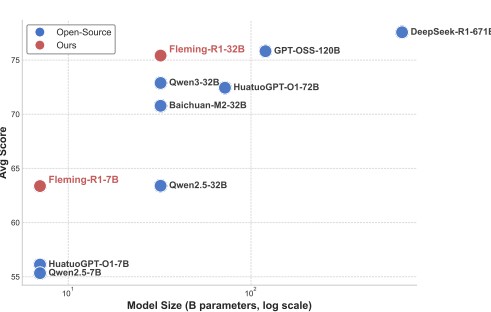

(a) MedXpertQA benchmark performance comparison across different models.

(b) Average benchmark performance comparison across different models.

Figure 1: Benchmark performance comparison across different models.

## 1 INTRODUCTION

While Large language models (LLMs) are increasingly applied to medicine, expert-level clinical reasoning remains a high-complexity, high-stakes frontier Liu et al. (2025b); Singhal et al. (2023; 2025); Moor et al. (2023). Clinical reasoning involves constructing extended, auditable chains of inference. These chains must integrate heterogeneous signals (such as history, physical exam, labs, and imaging) with evolving evidence-based guidelines, and weigh risks and benefits under uncertainty Joseph et al. (2025); Sun et al. (2025). Unlike general-domain tasks, success hinges on mapping

nuanced observations to pathophysiology and treatment principles, rather than just retrieval. A confident but incorrect answer is not merely suboptimal — it can be unsafe. Therefore, verifiability of reasoning (transparent steps that can be checked) is as central as aggregate accuracy Alufaisan et al. (2021); Coussement et al. (2024).

Despite encouraging results of LLMs on standardized clinical benchmarks Zuo et al. (2025); Jin et al. (2021); Pal et al. (2022); Jin et al. (2019); Wang et al. (2024; 2025a); Arias-Duart et al. (2025), current systems still struggle to produce transparent and reliable reasoning processesTurpin et al. (2023). In other words, models may output correct answers but fail to produce faithful, internally consistent chains of thought or maintain guideline concordance under paraphrase or case variations Lanham et al. (2023). When accuracy is measured against outcome-linked ground truth in realistic scenarios (e.g., acute abdominal syndromes), degradations become more apparent, often accompanied by overconfidence and non-transparent trajectories. These observations suggest that simply scaling parameters or naively optimizing final-answer accuracy is insufficient for clinical readiness.

We attribute the verifiability limitations of existing works to three key factors. First, existing data formulation is dominated by static QA pairs with sparse rationale supervision and limited coverage of long-tail entities (such as rare diseases, niche drugs, and atypical presentations). Such data formulation reduces exposure to multi-hop reasoning and trade-off analysis. Second, optimization objectives primarily reward final correctness, offering weak signals about where or why reasoning fails (such as dosing errors, unjustified diagnostic leaps, or guideline deviations). Third, curriculum and initialization lack structured guidance at cold start, producing fragile schemas that collapse on out-of-distribution or compositionally complex cases.

In this paper, we propose Fleming-R1, a model for expert-level medical reasoning that is verifiable, scalable, and parameter-efficient. Fleming-R1 comprises three mutually reinforcing components that align data design, reasoning capacity initialization, and reinforcement learning with checkable signals:

1. **Reasoning-Oriented Data Strategy (RODS).** We balance curated public medical QA corpora with knowledge-graph – guided synthesis from a Wikipedia-derived medical graph (over 100,000 entities) encoding relations among diseases, symptoms, laboratory tests, imaging findings, drugs, mechanisms, and contraindications. RODS explicitly emphasizes underrepresented diseases and drugs, and constructs reasoning-intensive items by sampling multi-hop paths (e.g., `symptom → pathophysiology → test → treatment`). Distractors are procedurally generated to be plausible-but-wrong via relation-preserving perturbations (e.g., competing diagnoses that share core features but diverge on discriminative labs), compelling models to articulate disambiguating evidence. The synthetic set is balanced against curated data to preserve realism while expanding long-tail coverage and compositional depth.

2. **Chain-of-Thought (CoT) cold-start.** We establish foundational reasoning policies by distilling reasoning trajectories from high-capacity teachers using pass@k-based selection with iterative refinement (backtracking, path exploration, and self-correction). Candidate trajectories are filtered by verifiable signals (consistency of intermediate calculations, unit correctness, alignment with guideline snippets) and by brevity/locality criteria (explicit statements of assumptions and uncertainties).

3. **Two-stage Reinforcement Learning from Verifiable Rewards (RLVR).** Using Group Relative Policy Optimization (Shao et al., 2024), Stage 1 consolidates core skills on moderate-difficulty cases with verifiable rewards: structured answer parsing, format checking. Stage 2 targets persistent failure modes via adaptive hard-sample mining to enhance reasoning capabilities when confronting challenging problems.

This paper makes the following contributions:

- We present Fleming-R1, a model that integrates RODS, CoT cold-start, and two-stage RLVR to address the problem of models generating final answers without providing a coherent reasoning process, thereby significantly enhancing its effectiveness in handling complex medical problems.
- We demonstrate strong parameter efficiency and scalability: the 7B-parameter variant surpasses 72B-class baselines on key medical benchmarks, while the 32B-parameter variant

achieves parity with closed-source state-of-the-art models (e.g., GPT-4o) across multiple benchmarks—together validating that our training regimen maximizes reasoning performance under tight parameter budgets.

- We release the model to facilitate reproducibility, compliance auditing, and collaborative advancement of medical AI research.

## 2 RELATED WORK

The deployment of Large Language Models (LLMs) in professional domains has advanced from foundational research to overcoming practical barriers Raza et al. (2025); Wang & Zhang (2024); Wang et al. (2025b); Li et al. (2024). Research focuses on three main directions: injecting domain-specific knowledge, adapting general reasoning, and optimizing decision-making with reinforcement learning. The medical field is a key area for these applications, where the robustness and verifiability of clinical reasoning are paramount. Our work addresses this core challenge.

Early medical LLMs were enhanced with specialized knowledge through techniques like supervised fine-tuning with medical knowledge graphs Kraljevic et al. (2021); Wang et al. (2023a). However, these models struggle with complex, multi-step clinical reasoning and often exhibit a disconnect between their knowledge reserves and practical application, a phenomenon described as "answer without justification" Aljohani et al. (2025). This gap is evident as LLMs still underperform compared to human clinicians in diagnostic tasks Hager et al. (2024), indicating that merely increasing model or data scale is insufficient. The primary challenge lies in embedding rigorous, verifiable medical reasoning processes.

To improve reasoning, general techniques like Chain-of-Thought (CoT) Wei et al. (2022); Kojima et al. (2022); Wang et al. (2023b) have been adapted for medicine Liu et al. (2024). Works such as HuatuoGPT-O1 have shown that explicit reasoning paths can improve performance on medical tasks Chen et al. (2025); Nori et al. (2023). Nonetheless, this approach faces significant hurdles, including the high cost of creating expert-verified medical CoT data and the tendency for generic reasoning paths to neglect the specific logical paradigms of clinical decision-making.

Reinforcement Learning (RL) offers another avenue, with a trend shifting from outcome-oriented (RLHF) Ouyang et al. (2022) to process-oriented optimization Liu et al. (2025a); Lai et al. (2025); Zhang et al. (2025). Recent innovations include dynamic verification systems with patient simulators to provide feedback, as seen in Baichuan-M2 Dou et al. (2025). A key limitation is that reward signals may not adequately target and correct logical errors within the reasoning chain. Furthermore, while various RL algorithms like PPO Schulman et al. (2017) and GRPO Shao et al. (2024) are being explored Chen et al. (2025); Lai et al. (2025); Shao et al. (2024), developing effective curriculum learning strategies to guide models through complex reasoning challenges remains an open area of research.

## 3 METHOD

As shown in Figure 2, the training pipeline of Fleming-R1 consists of three core stages: reasoning-oriented data strategy, reasoning capability cold start, and complex reasoning enhancement via reinforcement learning.

### 3.1 REASONING-ORIENTED DATA STRATEGY

To train a robust and reliable medical reasoning model, our multi-source data strategy integrates diverse data sources, filtering mechanisms, and synthetic data generation techniques. The data pipeline consists of three core components: (1) curation of diverse public medical question-answering datasets, (2) construction of large-scale synthetic data via automated knowledge discovery and topological sampling from a Wikipedia-derived medical knowledge graph, and (3) multi-stage data refinement including format validation, label correction, and difficulty-based stratification. This multi-source approach ensures comprehensive coverage of medical knowledge from both curated datasets and dynamically generated synthetic data.

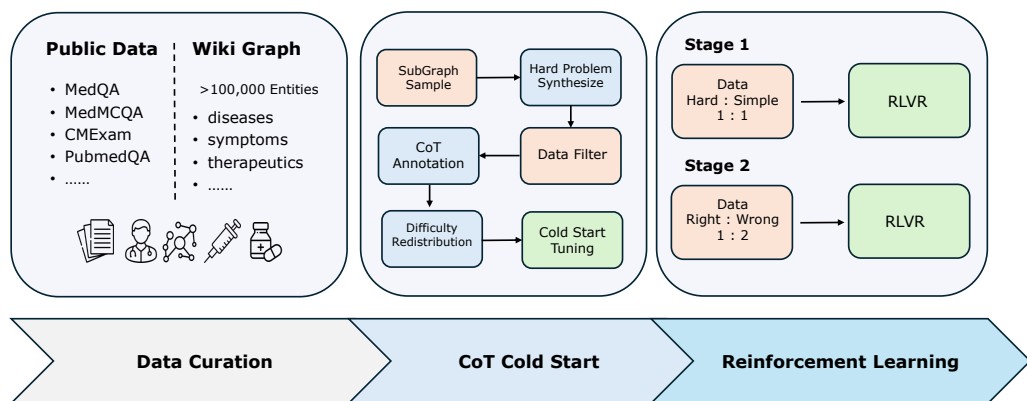

Figure 2: The overall training pipeline of Fleming-R1. This framework integrates a multi-source data strategy, reasoning capability cold start, and two-stage reinforcement learning with curriculum learning and GRPO for stable gains.

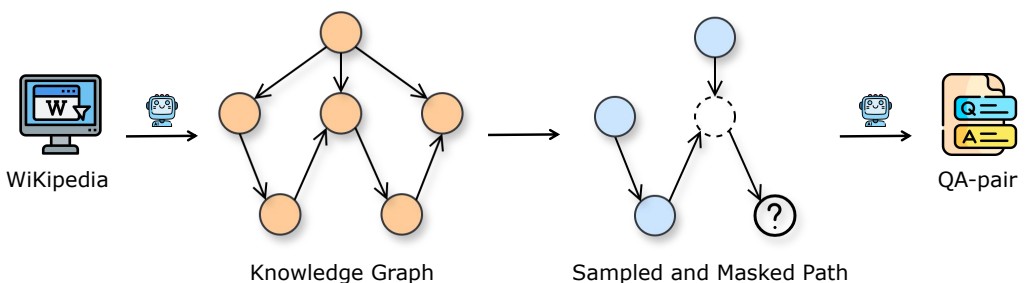

Figure 3: The pipeline for synthetic data generation. An autonomous agent discovers medical knowledge from Wikipedia to construct a knowledge graph. Subgraphs are then extracted via topological sampling and masked to create complex reasoning questions.

We begin by aggregating high-quality public medical QA datasets, including MedQA Jin et al. (2021), MedMCQA Pal et al. (2022), CMExam Liu et al. (2023), and PubMedQA Jin et al. (2019). These datasets provide a solid foundation of clinically relevant questions spanning a broad spectrum of medical domains, with explicit coverage across a comprehensive spectrum of medical domains—including diseases, symptoms, anatomy, physiology, diagnostics, therapeutics, drugs, and pathology—etc., ensuring comprehensive representation of medical knowledge essential for robust clinical reasoning. The MedQA and MedMCQA datasets offer challenging multiple-choice questions derived from medical licensing exams, providing a rigorous benchmark for factual knowledge and diagnostic reasoning. CMExam, a comprehensive Chinese medical exam dataset, ensures our model's capability extends to non-English medical contexts and diverse healthcare systems. PubMedQA, which contains questions derived from biomedical research abstracts, introduces a layer of complexity by requiring the model to understand and synthesize information from scientific literature, a crucial skill for evidence-based medicine.

To significantly expand the scope and depth of training data, we developed an autonomous knowledge discovery agent that systematically navigates Wikipedia to extract medical entities and their interrelations, constructing a large-scale medical knowledge graph comprising over 100,000 entities. The pipeline for this synthetic data generation is illustrated in Figure 3. This knowledge graph captures accurate, up-to-date, and verifiable medical information directly from a trusted source, mitigating the risk of hallucination during training. The agent performs entity linking and relation extraction to build a structured representation of medical knowledge, connecting concepts such as diseases, symptoms,

treatments, and anatomical structures. From this graph, we employ a topological sampling method to extract subgraphs representing coherent medical concepts or clinical scenarios. A key aspect of our sampling strategy is the deliberate focus on less common diseases and drugs. By prioritizing these underrepresented entities, we generate a higher proportion of challenging questions that require specialized knowledge and complex reasoning, thereby directly enhancing the model's ability to handle rare and difficult cases. By randomly masking portions of these subgraphs, we generate complex reasoning questions that challenge the model's ability to perform inference under partial information—a critical skill in real-world clinical decision-making. For instance, a question might present a patient's symptoms and lab results (the observed inputs) and ask for a diagnosis (the masked label), requiring the LLM to synthesize the evidence, weigh multiple hypotheses, and select the most appropriate diagnosis. This synthetic data generation process ensures both factual accuracy and pedagogical value, enabling the model to learn robust reasoning patterns grounded in real medical knowledge. It also allows us to create a vast number of unique training instances, particularly for rare conditions or complex interactions that are underrepresented in public datasets.

All collected and generated data undergo a multi-phase filtering and preparation pipeline. First, format-based filtering removes instances with structural anomalies such as duplicate answer options, malformed inputs, or encoding artifacts. Second, we implement a label accuracy verification step using a large language model as a validator. Specifically, any instance that fails to be correctly answered by a state-of-the-art LLM (e.g., GPT-4) across five independent trials is flagged for manual review to determine whether the labeling is incorrect. This step acts as a robust quality control mechanism, filtering out any erroneous or ambiguous data that could mislead the model. Additionally, sensitive information is systematically anonymized during preprocessing to ensure patient privacy and data safety. The final training dataset is constructed through deliberate data mixing, balancing the proportion of public and synthetic data to optimize model performance across knowledge breadth and reasoning depth. This mixing strategy is carefully tuned to prevent the model from overfitting to the patterns in synthetic data while still leveraging its benefits for enhancing reasoning capabilities.

Finally, we perform difficulty-level annotation using a large language model to classify each question into one of three tiers: *Easy*, *Moderate*, or *Difficult*. This classification is based on the cognitive and domain expertise demands of the question: *Easy* questions assess basic medical knowledge commonly known among practitioners; *Moderate* questions require detailed medical understanding or intermediate clinical reasoning; and *Difficult* questions demand advanced or specialized knowledge, complex multi-step inference, or familiarity with rare conditions. The difficulty-based bucketing strategy is integral to curriculum learning, enabling staged training from foundational concepts to complex diagnostic challenges, and supports targeted evaluation across different levels of complexity. This allows us to first stabilize the model on fundamental knowledge before progressively introducing more challenging problems, leading to a more robust and generalizable model.

## 3.2 Reasoning Capability Cold Start

To establish a robust foundation for advanced reasoning, we introduce a targeted cold start phase that directly imbues the base model with sophisticated reasoning behaviors. Rather than treating supervised fine-tuning as a conventional knowledge transfer step, we reframe it as a strategic cold start of reasoning patterns. Our approach centers on distilling expert-level reasoning trajectories from a high-capacity teacher model (e.g., GPT-OSS-120B) into the student model through a curated dataset of complex medical problems. For each query, we provide the base model with the question and its ground-truth answer, prompting the teacher model to generate a concise, logically structured Chain-of-Thought (CoT) that bridges the two. This ensures the reasoning is both accurate and pedagogically effective, focusing on essential inferential steps while avoiding extraneous detail.

To elevate the quality of reasoning further, we implement an iterative refinement protocol for the most challenging cases. The teacher model first generates an initial CoT, which is then evaluated against the ground truth. If the reasoning is incomplete or flawed, we initiate a refinement loop where the model revises its output using advanced strategies: (1) **Backtracking** to re-examine earlier assumptions, (2) **Path Exploration** to generate alternative hypotheses, and (3) **Self-Correction** to identify and fix logical errors. This meta-cognitive process produces final reasoning trajectories that reflect deep, reflective thinking with built-in error correction. By training on these high-quality, self-validated reasoning paths, the model internalizes the practice of "thinking before answering," a hallmark of expert clinicians. This cold start phase is not merely about learning facts but about

acquiring a robust reasoning framework, preparing the model for the subsequent stage of complex reasoning enhancement through reinforcement learning.

### 3.3 COMPLEX REASONING ENHANCEMENT VIA REINFORCEMENT LEARNING

Building upon the reasoning foundation established during the cold start, we introduce a reinforcement learning (RL) phase designed to amplify the model's complex reasoning capabilities. This stage moves beyond simple accuracy optimization, focusing instead on cultivating deep, resilient reasoning patterns through a dynamically adaptive training framework.

To refine the policy $\pi_\theta$, we employ Group Relative Policy Optimization (GRPO). This algorithm updates the policy by rewarding outputs that are better than the average of other candidate outputs generated for the same input. The objective is to minimize the following loss function:

$$\mathcal{L}_{\text{GRPO}} = -\mathbb{E}_{x \sim \mathcal{D}, \{y_i\}_{i=1}^k \sim \pi_\theta(\cdot|x)} \left[ \frac{1}{k} \sum_{i=1}^k \log \pi_\theta(y_i|x) \cdot A(x, y_i) \right] \tag{1}$$

The advantage function $A(x, y_i)$ is what distinguishes GRPO. For each input $x$, we first sample a group of $k$ candidate outputs, $\{y_1, y_2, \ldots, y_k\}$, from the current policy $\pi_\theta$. The advantage for a specific candidate $y_i$ is then computed relative to the average performance of this group:

$$A(x, y_i) = r(x, y_i) - \bar{r}_{G(x)} \tag{2}$$

Here, $r(x, y_i)$ is the total reward for the trajectory $y_i$. To mitigate the risk of reward hacking, our reward scheme is deliberately restricted to two criteria: correctness of the final answer and adherence to the required reasoning format. We deliberately exclude all other potential confounding factors—e.g., response length—from influencing the reward signal. The term $\bar{r}_{G(x)}$ is the group-level baseline, which is the average reward across all $k$ sampled candidates for the input $x$:

$$\bar{r}_{G(x)} = \frac{1}{k} \sum_{j=1}^k r(x, y_j) \tag{3}$$

By normalizing rewards within a group of contextually similar outputs, this baseline significantly reduces the variance of the gradient updates. This approach provides a more stable training signal and effectively encourages the model to discern and favor superior reasoning paths over other plausible alternatives.

Our RL framework follows a two-phase curriculum design. The first phase emphasizes the consolidation of fundamental reasoning skills through a balanced blend of Easy and Moderate difficulty questions. This promotes stable policy updates and steady learning progress. Once the model's performance plateaus—signaled by the emergence of reward sparsity—we transition to the second phase, which focuses on complex reasoning enhancement. Here, we introduce an adaptive hard sample mining strategy: the model is evaluated across the full dataset, and its repeated failures—particularly on Difficult questions requiring multi-step inference or specialized knowledge—are identified as high-priority training samples.

To address reward sparsity as the model improves, the second stage adopts an iterative curriculum learning approach that continuously refines the training distribution to target the model's current weaknesses. We use the model from the previous phase to detect reasoning errors and dynamically adjust the difficulty mix. Furthermore, we increase the number of rollouts during on-policy training to encourage broader exploration. This approach enables the acquisition of more sophisticated and robust reasoning strategies, leading to strong performance on challenging medical reasoning tasks.

## 4 EXPERIMENTS

This section presents the evaluation of our medical language model, detailing the benchmarks used, baseline models for comparison, and the experimental results.

Table 1: Main results on medical benchmarks. Our model sets new state-of-the-art performance on both 7B and 32B scales.

| Model | CareQA | JMED | Medbullets | MedMCQA | MedQA | MedXpertQA | PubMedQA | MMLU-Pro Biology | Health | Avg. |
|---|---|---|---|---|---|---|---|---|---|---|
| **> 100B** | | | | | | | | | | |
| DeepSeek-R1-671B | **93.68** | **66.50** | 79.87 | **80.40** | 92.93 | **37.59** | 76.00 | **90.24** | 80.93 | **77.57** |
| GPT-OSS-120B | 91.25 | 64.70 | **81.54** | 75.09 | 90.97 | 34.73 | **78.20** | 89.96 | 75.92 | 75.82 |
| **10B–100B** | | | | | | | | | | |
| **Fleming-R1-32B** | **90.41** | 68.70 | **76.51** | 74.52 | **89.32** | **30.33** | **80.40** | **90.93** | **77.63** | **75.42** |
| Qwen3-32B | 88.29 | **69.30** | 71.81 | 72.51 | 86.96 | 26.04 | 77.00 | 88.56 | 75.55 | 72.89 |
| HuatuoGPT-O1-72B | 87.69 | 61.70 | 72.48 | **76.02** | 88.30 | 24.65 | 79.80 | 86.61 | 74.82 | 72.45 |
| Baichuan-M2-32B | 86.05 | 64.00 | 70.81 | 69.81 | 88.22 | 23.39 | 75.20 | 83.96 | 75.55 | 70.78 |
| GPT-OSS-20B | 87.08 | 60.40 | 71.48 | 68.78 | 85.55 | 26.45 | 77.40 | 85.50 | 72.00 | 70.51 |
| Qwen2.5-32B | 81.55 | 66.50 | 48.99 | 71.56 | 71.56 | 13.63 | 73.60 | 82.01 | 68.22 | 63.40 |
| **< 10B** | | | | | | | | | | |
| **Fleming-R1-7B** | **77.28** | **59.60** | **57.05** | **64.16** | **75.10** | **19.14** | **78.60** | **74.76** | **64.67** | **63.37** |
| HuatuoGPT-O1-7B | 72.00 | 52.70 | 41.61 | 62.11 | 66.30 | 10.94 | 64.46 | 74.34 | 60.51 | 56.12 |
| Qwen2.5-7B | 70.56 | 59.20 | 42.95 | 55.89 | 59.86 | 11.96 | 74.00 | 72.38 | 52.08 | 55.43 |

## 4.1 EVALUATION SETTINGS

Our benchmarks and baselines are detailed in Appendix B and C.

We selected Qwen2.5-7B (Team, 2024) as the base model for Fleming-R1-7B, and Qwen3-32B (Yang et al., 2025) as the base model for Fleming-R1-32B. The Fleming-R1-7B model underwent a full training process including CoT cold-start and RLVR training. In contrast, since Qwen3 already possesses substantial reasoning capabilities, the Fleming-R1-32B model only received RLVR training.

## 4.2 EXPERIMENTAL RESULTS

We evaluate on nine medical benchmarks. Table 1 reports per-task accuracy and the macro average ("Avg.").

**Main results by model size.** At the < 10B scale, Fleming-R1-7B attains the best average (63.37%), outperforming HuatuoGPT-O1-7B (56.12%) and Qwen2.5-7B (55.43%) by +7.25 and +7.94 percentage points (pp), respectively. It ranks first on all reported tasks within this size class (e.g., CareQA 77.28%, MedMCQA 64.16%, MedQA 75.10%, PubMedQA 78.60%, MedXpertQA 19.14%). Notably, despite being 7B, it surpasses the 32B Qwen2.5 model on several benchmarks (e.g., MedBullets, MedQA, MedXpertQA, PubMedQA), indicating strong parameter efficiency.

Within 10B–100B, Fleming-R1-32B achieves the highest average (75.42%), ahead of Qwen3-32B (72.89%), HuatuoGPT-O1-72B (72.45%), Baichuan-M2-32B (70.78%), and GPT-OSS-20B (70.51%) by +2.53, +2.97, +4.64, and +4.91 pp, respectively. It leads on 7/9 tasks at this scale—CareQA (90.41%), MedBullets (76.51%), MedQA (89.32%), MedXpertQA (30.33%), PubMedQA (80.40%), and both MMLU-Pro subsets (Biology 90.93%, Health 77.63%)—while remaining close on the two remaining tasks (JMED 68.70% vs. 69.30% for Qwen3-32B; MedMCQA 74.52% vs. 76.02% for HuatuoGPT-O1-72B).

**Against larger models.** Although trained at 32B, Fleming-R1 approaches the > 100B tier. Its average (75.42%) is within 2.15 pp of DeepSeek-R1-671B (77.57%) and within 0.40 pp of GPT-OSS-120B (75.82%). Moreover, Fleming-R1-32B surpasses GPT-OSS-120B on 4/9 tasks, including JMED (68.70% vs. 64.70%), PubMedQA (80.40% vs. 78.20%), and both MMLU-Pro subsets (Biology 90.93% vs. 89.96%; Health 77.63% vs. 75.92%). These head-to-head results highlight strong generalization and reasoning capabilities relative to substantially larger systems.

## 4.3 ABLATION ANALYSIS

Table 2 disentangles the contribution of each training stage for Fleming-R1 at 7B and 32B.

**7B.** Starting from the Non-Inference baseline (Avg 55.4%), adding the CoT cold start yields a clear gain to 58.5% (+3.1 pp), indicating that explicit early-stage reasoning scaffolds benefit downstream medical QA. Introducing RLVR (Stage 1) further lifts performance to 61.2% (+5.8 pp over Base). Our

Table 2: Ablation of training stages for Fleming-R1 at 7B and 32B. Numbers are accuracy (%). $\Delta$Avg is the absolute gain over the corresponding Base within the same size. Best results per size in **bold**.

| Model Variant | | CareQA | JMED | Medbullets | MedMCQA | MedQA | MedXpertQA | PubMedQA | MMLU-Pro | | Avg. | $\Delta$Avg |
| Size | Variant | | | | | | | | Biology | Health | | |
|---|---|---|---|---|---|---|---|---|---|---|---|---|
| 7B | Base | 70.6 | 59.2 | 43.0 | 55.9 | 59.9 | 12.0 | 74.0 | 72.4 | 52.1 | 55.4 | +0.0 |
| | +COT Cold Start | 72.2 | 54.4 | 52.7 | 58.5 | 67.2 | 16.1 | **78.6** | 69.9 | 57.0 | 58.5 | +3.1 |
| | +RL Stage 1 | 75.9 | **59.6** | 53.7 | 61.5 | 69.4 | 17.2 | 77.4 | **74.8** | 61.3 | 61.2 | +5.8 |
| | +RL Stage 2 | **77.3** | **59.6** | **57.1** | **64.2** | **75.1** | **19.1** | 78.6 | **74.8** | **64.7** | **63.4** | **+7.9** |
| 32B | Base | 88.3 | 69.3 | 71.8 | 72.5 | 87.0 | 26.0 | 77.0 | 88.6 | 75.6 | 72.9 | +0.0 |
| | +RL Stage 1 | 90.0 | **70.1** | 73.5 | 74.0 | 88.4 | 27.7 | 78.8 | **91.2** | 76.9 | 74.5 | +1.6 |
| | +RL Stage 2 | **90.4** | 68.7 | **76.5** | **74.5** | **89.3** | **30.3** | **80.4** | 90.9 | **77.6** | **75.4** | **+2.5** |

full two-stage regimen—which couples RLVR with curriculum learning and adaptive hard-sample mining—delivers the best 7B result at $63.4\%$ (+7.9 pp). Improvements are broad-based rather than benchmark-specific: e.g., MedQA +15.2 pp ($59.9 \rightarrow 75.1$), MedBullets +14.1 pp ($43.0 \rightarrow 57.1$), MedMCQA +8.3 pp ($55.9 \rightarrow 64.2$), MedXpertQA +7.1 pp ($12.0 \rightarrow 19.1$), and MMLU-Pro (Health) +12.6 pp ($52.1 \rightarrow 64.7$). These trends suggest that Stage 2 effectively targets persistent failure modes and consolidates clinical reasoning under distributional stress.

**32B.** Given the stronger innate reasoning of the 32B model, we omit CoT cold start and focus on RLVR. The Base reaches $72.9\%$, RL Stage 1 improves to $74.5\%$ (+1.6 pp), and our full two-stage schedule attains $75.4\%$ (+2.5 pp). The largest per-task gains arise on MedBullets (+4.7 pp), MedXpertQA (+4.3 pp), PubMedQA (+3.4 pp), and MedQA (+2.3 pp), alongside steady advances on MMLU-Pro Biology/Health (+2.3/+2.0 pp). While JMED exhibits a minor fluctuation ($-0.6$ pp), the overall average increases monotonically across RL stages, indicating that targeted optimization on hard cases sharpens the model's already-strong reasoning.

**Ablation Summary.** Across both 7B and 32B settings, the two-stage RLVR consistently improves accuracy while scaling with model capacity. At 7B, it yields a +7.9 pp gain over the Base (Avg 55.4→63.4), with broad improvements on MedQA (+15.2 pp), MedBullets (+14.1 pp), MedMCQA (+8.3 pp), MedXpertQA (+7.1 pp), and MMLU-Pro (Health) (+12.6 pp). At 32B, it adds +2.5 pp over the Base (72.9→75.4), with notable gains on MedBullets (+4.7 pp), MedXpertQA (+4.3 pp), PubMedQA (+3.4 pp), and steady advances on MMLU-Pro Biology/Health (+2.3/+2.0 pp), despite a minor dip on JMED (–0.6 pp). These results support our design: establish foundational reasoning early and then apply curriculum-guided RL on hard cases to eliminate residual errors and strengthen clinical reasoning robustness.

### 4.4 ANALYSIS OF REASONING CAPABILITIES

We evaluate clinical reasoning on MedXpertQA, a rigorously curated expert-level medical benchmark. Compared with prior medical QA suites, MedXpertQA increases difficulty via specialty-board style items, rich clinical contexts (e.g., patient records and exam results), leakage mitigation through data synthesis, and multi-round expert review, thereby stressing multi-hop, verifiable reasoning rather than shallow pattern matching. Figure 4 and Figure 1 summarizes our results ("*" from our runs; "+" from the official leaderboard).

**7B scale.** Fleming-R1-7B attains $19.14\%$ on MedXpertQA, substantially ahead of comparable 7B baselines (e.g., Qwen2.5-7B-Instruct $11.96\%$), and even surpasses some much larger general models (e.g., Qwen2.5-72B $18.90\%$). This highlights the parameter efficiency of our training recipe—CoT cold start plus two-stage RLVR—under expert-level clinical difficulty.

**32B scale.** Fleming-R1-32B reaches $30.33\%$, achieving near parity with GPT-4o at $30.37\%$ (absolute gap **0.04** points; $\approx 0.13\%$ relative) while remaining fully open-source. Among $\leq$32B open models, it establishes a new strong baseline (e.g., Qwen3-32B $26.04\%$, Baichuan-M2-32B $23.39\%$), demonstrating that our approach closes most of the remaining gap to leading closed-source systems on complex medical reasoning.

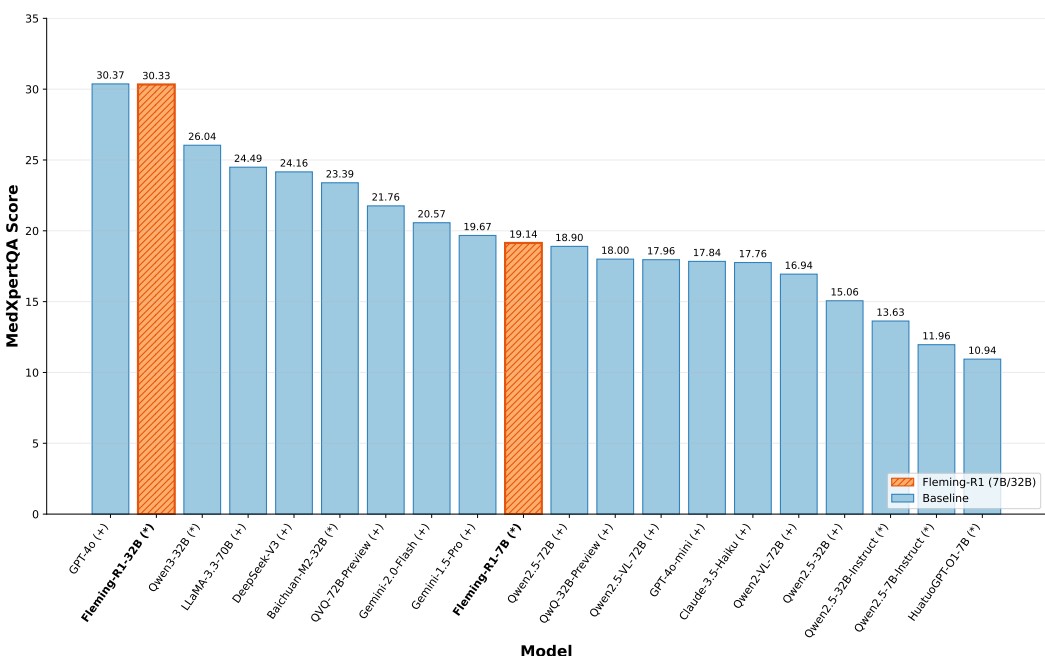

Figure 4: Comparison on MedXpertQA across models. "*" indicates results from our runs; "+" from the official leaderboard. Fleming-R1 achieves near–GPT-4o performance at 32B and leads among 7B models, underscoring parameter efficiency and expert-level clinical reasoning under a high-difficulty benchmark.

**Discussion.** Taken together, these results on MedXpertQA—a benchmark expressly designed to assess expert medical reasoning—indicate that our framework does more than memorize facts: the CoT cold start builds multi-source reasoning priors, and the curriculum-driven two-stage RLVR (with adaptive hard-sample mining) systematically attacks persistent failure modes. The outcome is a scalable, parameter-efficient improvement in clinical reasoning, from 7B (strong gains over peers) to 32B (GPT-4o-level performance) within an open-source paradigm.

## 5 CONCLUSION

We present Fleming-R1, a model for expert-level medical reasoning that targets core limitations of current LLMs in clinical settings. Our training framework combines three complementary components: (i) a reasoning-oriented data strategy, (ii) a Chain-of-Thought (CoT) cold start that lays a foundation for structured inference, and (iii) a two-stage RLVR regimen with curriculum learning and GRPO to deliver stable gains in correctness and consistency.

Empirically, Fleming-R1 achieves strong, scale-consistent improvements on expert-level evaluation. On MedXpertQA—a challenging benchmark spanning 4,460 items across 17 specialties and 11 body systems—our 7B model attains state-of-the-art performance among comparable models and even surpasses larger systems, evidencing substantial parameter efficiency. The 32B model reaches 30.33%, essentially matching GPT-4o (30.37%) while exceeding open baselines, and delivers competitive results across a comprehensive medical suite. These outcomes validate our design: establish broad, structured reasoning priors early, then refine them via verifiable, curriculum-guided RL to reduce persistent error modes.

We release our model as an open resource to support transparent, reproducible, and auditable research in clinical AI. In addition to helping advance medical reasoning capabilities, Fleming-R1 aims to facilitate the verification of model behavior, support compliance auditing, and promote safer deployment in high-stakes medical settings.

ETHICS STATEMENT

We acknowledge the ICLR Code of Ethics and confirm that our work adheres to its principles. Our research does not involve human subjects or personally identifiable information. The datasets we used are publicly available, and we strictly followed their licenses and usage policies. We have carefully considered potential risks of privacy leakage, bias, or unfairness. We believe our findings contribute positively to the research community without foreseeable harmful applications.

REPRODUCIBILITY STATEMENT

We are firmly committed to ensuring the reproducibility of our research. To this end, we have provided a comprehensive description of our methodology in Section 3 and a thorough account of our experimental configurations and settings in Section 4, Appendix B and C. Furthermore, to facilitate the verification of our results and to encourage future research in this area, we will make our models publicly available upon the publication of this work.

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

# APPENDIX

## A THE USE OF LARGE LANGUAGE MODELS (LLMS)

In accordance with the ICLR policy on LLM usage, we hereby disclose that Large Language Models (LLMs) were only used as auxiliary tools for language polishing and minor grammatical improvements in the writing process. They were not involved in research ideation, experimental design, implementation, data analysis, or the generation of scientific content. The authors take full responsibility for the content of this paper.

## B DETAILS OF BASELINES

We compare our model with strong general-purpose and medical-domain baselines, focusing on state-of-the-art systems that represent the current frontier in their respective areas (e.g., DeepSeek-R1 (Guo et al., 2025), Baichuan-M2 (Dou et al., 2025), Qwen3 (Yang et al., 2025), HuatuoGPT-O1 (Chen et al., 2025)). Table 3 summarizes these baselines with parameter counts and whether they include inference-time reasoning.

## C DETAILS OF BENCHMARKS

To assess our model's capabilities, we evaluate it on a comprehensive suite of eight medical benchmarks, detailed below and summarized in Table 4. We report accuracy on the standard close-ended (MCQ) splits. We instruct the model to enclose the answer choices within the <answer></answer> tokens to facilitate accurate extraction of the response.

- **MedXpertQA (Text)** Zuo et al. (2025): Expert-level medical QA with 4,460 questions spanning 17 specialties and 11 body systems, provided in text-only and multimodal subsets to assess advanced reasoning under clinically realistic settings. We used the text-only subset.
- **MedQA (USMLE)** Jin et al. (2021): Multiple-choice questions collected from professional medical board exams (commonly referenced via the USMLE split), widely used to measure broad medical knowledge and diagnostic reasoning.

Table 3: Baseline models and their sizes and reasoning capabilities.

| Model | Parameters | Reasoning Capability |
|---|---|---|
| DeepSeek-R1 | 671B | Inference |
| GPT-OSS | 20B, 120B | Inference |
| Baichuan-M2 | 32B | Inference |
| Qwen3 | 32B | Inference |
| HuatuoGPT-O1 | 7B, 72B | Inference |
| Qwen2.5 | 7B, 32B | Non-inference |

"Reasoning Capability" indicates whether the model supports inference-time (test-time) reasoning mechanisms.

Table 4: Benchmarks used in our evaluation.

| Benchmark | Data Source | Answer Format | Test Dataset Size |
|---|---|---|---|
| MedXpertQA (Text) | Examination | 4-option MCQs | 2,450 |
| MedQA (USMLE) | Examination | 4-option MCQs | 1,273 |
| MedMCQA | Examination | 4-option MCQs | 4,183 |
| MMLU-Pro (Biology) | Examination | 10-option MCQs | 717 |
| MMLU-Pro (Health) | Examination | 10-option MCQs | 818 |
| CareQA | Examination | 4-option MCQs | 5,621 |
| JMED | Hospital | 21-option MCQs | 1,000 |
| PubMedQA | Literature | 3-option MCQs | 1,000 |

- **MedMCQA** Pal et al. (2022): Large-scale MCQ benchmark sourced from AIIMS and NEET PG entrance exams (194k items across 21 subjects), designed to stress multi-subject medical knowledge and reasoning.

- **PubMedQA** Jin et al. (2019): Biomedical QA where each item asks a research question answered as *yes/no/maybe* from the corresponding PubMed abstract; includes a 1k expert-labeled test set.

- **MMLU-Pro (Biology)** Wang et al. (2024): Biology subset of MMLU-Pro, which increases difficulty and robustness over MMLU by using ten-option MCQs and more reasoning-centric items.

- **MMLU-Pro (Health)** Wang et al. (2024): Health subset of MMLU-Pro under the same ten-option, reasoning-oriented setting.

- **JMED** (Wang et al., 2025a): A clinical-practice evaluation set constructed from anonymized doctor–patient dialogues at JD Health Internet Hospital. The evaluation split is cast as 21-option MCQs (including a "None of the above" choice) to reflect ambiguity in real consultations and enable continuous updates.

- **CareQA** Arias-Duart et al. (2025): A newly released benchmark derived from Spain's Specialized Healthcare Training (FSE/MIR) exams (2020–2024). It includes a close-ended MCQ set (5,621 items across medicine, nursing, biology, chemistry, psychology, and pharmacology) and an English open-ended variant created via controlled rephrasing and human review.

