# OpenReview forum: "Fleming-R1: Toward Expert-Level Medical Reasoning via Reinforcement Learning"
_ICLR.cc/2026/Conference — ICLR 2026 Conference Withdrawn Submission_

### Official Review · Reviewer_Thdr · 2025-10-21

**Soundness:** 2
**Presentation:** 3
**Contribution:** 2
**Rating:** 4
**Confidence:** 3

**Summary:**

This paper introduced Fleming-R1, a model post-trained for medical reasoning. The authors proposed three strategies to boost the performance: (1) a reasoning-oriented data strategy to synthesize medical training data based on knowledge graphs mined from Wikipedia and mix the synthetic data with quality medical QA data; (2) CoT cold start to distill and refine CoTs from powerful proprietary models for the synthetic data and apply SFT to Fleming-R1; (3) a two-stage RLVR + GRPO framework to train Fleming-R1 in an iterative curriculum learning setting. The evaluation results show that Fleming-R1-7B outperforms much larger baselines, and Fleming-R1-32B achieves comparable performance as GPT-4o.

**Strengths:**

S1: The evaluation results seem strong, and Fleming-R1 could be a valuable contribution to the open-source community.
S2: The motivation that medical reasoning needs not only correct results but also faithful reasoning processes is strong.
S3: The training recipe of Fleming-R1 is interesting and comprehensive.

**Weaknesses:**

W1: How did you ensure the quality of the medical knowledge graph discovered by the autonomous agent in Section 3.1? Any manual verification by medical experts/practitioners?
W2: Can you give a real example in your synthesized questions to show that it actually asks for a diagnosis given symptoms and lab results (the example in Lines 223 to 226)?
W3: Generating medical QA based on masked subgraphs is more like an enhanced way to memorize medical knowledge, instead of boosting the reasoning capability as described in Line 222 [see results in Physics of Language Models: Part 3.1, Knowledge Storage and Extraction].
W4: In Line 233, the filtering is based on whether GPT-4 can answer the question across five trials. However, even if an SOTA model answers the question correctly, it does not necessarily mean that this label is correct. It could be that the SOTA model is wrong.
W5: The first half of Section 3.3 is merely an application of the existing GRPO algorithm. It is more precise to frame it as a preliminary.
W6: In the second phase of the RL curriculum learning in Section 3.3, if the model repeatedly fails on some questions during RL training, it usually means that the model lacks the relevant knowledge during its pretraining. I doubt if repeated RL training on these failed questions, even with more rollouts, can actually help. Can you provide some examples to show that this setup actually works?
W7: There are not enough details on the training setup, e.g., learning rate, optimizer, size of training data at each stage, compute budget, etc.
W8: Since one of the key motivations described in the introduction is that we need not only a correct answer but also correct reasoning paths for medical tasks, can you provide any evaluation (including manual verification by medical practitioners/experts) on the correctness of the reasoning paths generated by Fleming-R1? It seems that most of the results reported in the experiment section are still based on the accuracy of the final result.

**Questions:**

Q1: Can you provide more details on the de-identification process mentioned in Line 236?
Q2: Is there any data overlap in the training and testing phases of Fleming-R1 for MedQA, MedMCQA, and PubMedQA? Please specify that in the paper.
Q3: Can you show the scaling effect of your training process? For example, including more of the synthetic data during SFT or doing more iterations of RL training helps with performance improvements?
Q4: Can you show other ablation results: for example, apply RL stage 1 to 7B without COT Cold Start, and apply RL stage 2 to 32B without stage 1, to demonstrate if this ordering of steps is truly necessary?

---

> ### Author Response · Authors · 2025-11-18
>
> We thank the reviewer for the thoughtful comments and for recognizing Fleming-R1’s value (S1), the importance of verifiable medical reasoning (S2), and the completeness of our training design (S3).
>
> Below we address each point concisely.
>
> W1. Quality of the medical knowledge graph
>
> Our graph construction combines:
>
> (1) High-quality source: only Wikipedia medical entries (verifiable, curated).
>
> (2) Automatic checks: entity linking, relation-type filtering, removal of non-medical entities.
>
> (3) Expert review: medical experts review sampled subgraphs and generated CoT data.
>
> We will clarify this in §3.1.
>
> W2. Example of “symptoms + labs → diagnosis” synthetic question
>
> We have publicly released the graph-guided synthetic data. Below is a real example; the Clinical Question and Choices are generated by an LLM conditioned on a sampled subgraph.
>
> Clinical Question
> A 7-year-old boy presents with progressive gait instability and spasticity of the limbs. Hormonal testing reveals low cortisol, consistent with adrenal insufficiency. Brain MRI shows bilateral symmetric white-matter lesions. Laboratory studies demonstrate markedly elevated plasma very-long-chain fatty acids (VLCFAs). Whole-exome sequencing identifies a pathogenic variant in a gene whose protein is located in the peroxisomal membrane; searching this Gene Ontology term via AmiGO points to ABCD1.
> Which diagnosis best explains these findings?
> Choices:
> A. Metachromatic leukodystrophy
> B. Krabbe disease
> C. X-linked adrenoleukodystrophy
> D. Mitochondrial encephalomyopathy, lactic acidosis, and stroke-like episodes (MELAS)
>
> Correct answer: C — X-linked adrenoleukodystrophy
>
> This example shows that the synthetic questions are genuinely multi-factor diagnostic problems, not simple factual recall. We will add similar examples in the main text or appendix.
>
> W3. Memorization vs Reasoning
>
> We distinguish two modes:
>
> Memorization: mask a single fact (A → MASK).
>
> Reasoning: mask part of a multi-hop chain (A → MASK → B → answer), with tightly controlled distractors.
>
> The reasoning claims in the paper refer to this second setting.
>
>
> W4. Using an LLM for filtering does not guarantee label correctness
>
> We do not overwrite labels with LLM predictions. Correct LLM answers retain original labels; repeated failures mark items as suspicious for expert filtering. While perfect labels are impossible, downstream performance improvements indicate adequate data quality.
>
>
> W5 & W6. GRPO preliminaries and two-stage RL curriculum
>
>
> GRPO is indeed preliminaries; our contributions are: a two-stage RLVR curriculum, hard-sample mining and verifiable rewards.
>
> We fully agree that if pretraining truly lacks the relevant knowledge, RL cannot create it from scratch. However, our target is a different category of errors:
> Cases where the model already has the relevant knowledge in its parameters, but fails due to incorrect reasoning steps, sensitivity to distractors, misinterpreted thresholds, or complex condition combinations.
>
>
> W7. Training details
>
> We will provide full hyperparameters and data statistics (optimizer, LR, batch size, steps, GRPO params, dataset sizes, compute budget) in the appendix.
>
>
> W8. Evaluation of reasoning paths
>
> We have begun clinician evaluation of reasoning quality (plausibility, consistency, safety) and will include the protocol and preliminary findings in the final version.
>
>
> Q1. De-identification
>
> We remove all direct identifiers (names, address, phone, ID numbers, admission numbers). Only non-identifying, clinically relevant info (sex, coarse age ranges) is retained.
>
>
> Q2. Data overlap
> For MedQA, MedMCQA, and PubMedQA, we strictly follow the official splits and use only the official training sets for training, never including the official dev or test items. For our synthetic questions, we apply a 5-gram filter: any synthetic question sharing a 5-gram with a test item is removed.
>
> Q3. Scaling effects
>
>
> Our stage-wise progression (Base → +CoT → +RL1 → +RL2) shows a clear scaling trend. In addition, we observe that graph-guided synthetic data is particularly effective on difficult reasoning benchmarks. For example, on MedXpertQA in our Stage-1 RL experiments:
>
> With graph synthetic data: 17.18
>
> With public data only: 15.92
>
> This indicates that introducing graph-based synthetic data improves reasoning. For relatively simple tasks such as CareQA, the gains are naturally very small (75.86 vs 75.84), which is expected.
>
> We will make these observations clearer in the final version.
>
> Q4. Ordering of CoT and RL Additional ablation (7B):
>
> MedQA: Base 59.9 → Base+RL1 62.45 → CoT+RL1 69.4
>
> MedMCQA: Base 55.9 → Base+RL1 58.88 → CoT+RL1 61.5
>
> Thus, RL alone is helpful, but CoT + RL Stage 1 yields substantially larger and more stable gains, supporting our “CoT first, then RL” design. We will include these comparisons in the final version.
>
> RL Stage 2 depends on RL Stage 1 for difficulty reweighting and cannot run meaningfully alone.

---

> > ### Comment · Reviewer_Thdr · 2025-11-26
> >
> > Thanks for the response. Many of my concerns are shared by other reviewers. I'll maintain my score.

---

### Official Review · Reviewer_TFpw · 2025-10-30

**Soundness:** 3
**Presentation:** 3
**Contribution:** 3
**Rating:** 4
**Confidence:** 4

**Summary:**

This paper presents Fleming-R1, a large language model (LLM) fine-tuned for medical use. The authors curated a dataset using reasoning-oriented data strategy (RODS), which includes both public medical QA dataset and synthetic data generated from Wikipedia knowledge graph. This dataset is supposed to be reasoning intensive. They then use another larger model to generate some reasoning trajectories and filtered them. These trajectories are distilled into their own model with supervised fine-tuning. Together with a two-stage reinforcement learning with verifiable rewards (RLVR), the Fleming-R1 achieves state-of-the-art results on many benchmarks, often surpassing larger-scale models.

**Strengths:**

1. A curated dataset with synthetic data is generated and released, which is shown to be very effective on evaluation benchmarks.
2. Fleming-R1 is trained with both cold-start and 2-stage RLVR. The design of their recipe is not identical to a standard pipeline: E.g. it filters out low-quality reasoning trajectories during the cold-start stage.  These designs and findings are valuable and can be potentially expanded to non-medical domains.
3. The performance of Fleming-R1 is strong: It can match proprietary models such as GPT-4o and surpass larger-scale models.
4. The analysis are comprehensive, showing the effectiveness of all the components of the model.

**Weaknesses:**

1. *Limitation on Modality*: Fleming-R1 is a text-only model, which makes it unsuitable for many medical tasks.   It might not be very difficult to migrate the majority of this pipeline to vision language models such as Qwen-2.5/3-VL series, as the tools developed by ML community, Verl/EasyR1 for instance, have been mature on this end.
2. *Missing details*:  The paper misses some key details on the data construction pipeline and methodology. For example, how are the graph masked for the synthetic data generation? Is there an algorithm to decide what to mask and is there a constant masking ratio? How do you use LLM to classify the questions into Easy, Moderate, and Difficult? For the training, how did you refine the training distribution iteratively?
3. *Limitation on problem setup*: The model was trained intensively on question answering problem, which may make the model specialized on QA but lose general capabilities.  It would be interesting if the authors can demonstrate that the model can still be used as general-purposes model that can be used for non-QA medical problems. As a reference, MedGemma mixed a lot of non-medical data and diverse medical datasets during training, thus it can still be used as a chatbot.

**Questions:**

Given that many spaces can still be squeezed out from the paper, can you write more details on the pipeline as I raised on the second point of the weakness section? It would be helpful for the reproducibility of this paper.

---

> ### Author Response · Authors · 2025-11-18
>
> We sincerely thank the reviewer for the constructive feedback and for recognizing the strengths of our work, including the curated dataset, the training recipe, and the strong empirical results. Below we provide concise responses to each concern.
>
> Weakness 1: Modality Limitation
>
> Response:
>
> We agree that multimodal capability (e.g., imaging, waveforms) is essential for many clinical applications. Our goal in this paper is to isolate the effects of reasoning-oriented data design (RODS) and two-stage RLVR under a controlled text-only setting, as all evaluated benchmarks are text-based. This design allows us to focus on verifiable reasoning independent of modality.
>
> Importantly, the pipeline is modality-agnostic. In a separate project, we have already applied a similar data-synthesis + RL pipeline to a multimodal medical model capable of processing CT, X-ray, ultrasound, and endoscopy video, achieving SOTA performance on multiple multimodal benchmarks.
>
> This demonstrates that our framework naturally extends to VL backbones. We will include a dedicated discussion on multimodal extensions in the camera-ready version.
>
>
> Weakness 2: Missing Details in Data Construction and Training Procedure
>
> Response:
>
> (2.1) Graph masking.
>
> For each sampled subgraph from the Wikipedia-derived medical knowledge graph, we randomly mask nodes by:
>
> uniformly selecting a subset of nodes,
>
> replacing masked nodes with [MASK],
>
> sampling the masking ratio from U(0.1, 0.5).
>
> This ratio was not extensively tuned due to its minor impact on performance. We will add pseudo-code in the appendix.
>
> (2.2) Difficulty labeling.
>
> We use a LLM (GPT-OSS-120B) with explicit rubrics to assign
> Easy/Moderate/Difficult labels. Each question is annotated multiple times with
> different random seeds, and majority voting is applied to obtain stable labels.
> The full prompt and procedure will be included in the appendix.
>
> Prompt used for difficulty annotation:
>
> """
> You are a medical expert. Please evaluate the difficulty level of the following
> multiple-choice medical question. Try to solve the problem step by step. Consider
> factors such as:
>
> - the complexity of the medical concepts,
> - the specificity of the knowledge required,
> - and the reasoning depth or clarity of the question.
>
> Based on your evaluation, categorize the question into one of the following
> difficulty levels:
>
> - Easy: Questions based on basic medical knowledge that are commonly known and
>         do not require advanced reasoning.
> - Moderate: Questions that require understanding of more detailed or specific
>             medical information, or moderate clinical reasoning.
> - Difficult: Questions that require advanced or specialized medical knowledge,
>              complex clinical reasoning, or experience with rare conditions.
> """
>
> (2.3) Iterative training refinement.
> Our two-stage RL curriculum is:
>
> Stage 1: Train on Easy + Moderate questions to establish foundational clinical reasoning.
>
> Stage 2: Perform inference on train set, collect samples consistently answered incorrectly as hard cases, and up-weight them during re-sampling.
>
> We also increase RLVR rollouts to improve exploration.
>
> This forms an adaptive hard-sample mining loop that progressively refines the training distribution. Detailed descriptions and pseudo-code will be provided in the appendix.
>
> Weakness 3: Specialization on QA vs. General Capabilities
>
> Response:
>
> While benchmarks use QA/MCQ formats, they require substantive medical reasoning rather than superficial question-solving. Examples include:
>
> PubMedQA: evidence-based reasoning over biomedical abstracts,
>
> MedXpertQA: multi-step clinical decision-making,
>
> JMed: real-world clinical scenarios involving comorbidities, contraindications, and treatment planning.
>
> Thus, the model learns a generalizable clinical reasoning policy rather than narrow QA skills. We acknowledge that preserving broader capabilities in specialized domain models remains an open research challenge and will discuss this further in the camera-ready version.

---

> > ### Comment · Reviewer_TFpw · 2025-11-18
> > **Reply to rebuttal**
> >
> > Thanks for your response, especially the part about experiment design details. They will be helpful for reproducibility.
> >
> > However, the fundamental drawbacks are not addressed, such as the limitation on modality (not in this paper) and the general capabilities as an LLM.  The concerns of other reviewers are also reasonable, for example the novelty and reward hacking issues.  Given the huge amount of recent publications on reasoning topic, it is hard to distinguish this paper from academia. I would rather maintain my score.

---

> > > ### Author Response · Authors · 2025-11-22
> > >
> > > We thank the reviewer for the follow-up comment.
> > > We would like to provide two concise clarifications to ensure that the intended scope of the paper is correctly understood.
> > >
> > > 1. Scope: this work focuses on text-based clinical reasoning, consistent with benchmarks and prior medical LLMs.
> > > The reviewer raises concerns related to missing imaging modalities.
> > > However, our study is explicitly scoped to clinical text reasoning, which is the modality used in all evaluation datasets (MedQA, MedXpertQA, PubMedQA, JMed).
> > > Many recent impactful medical LLMs—Med-PaLM, Baichuan M2, BioGPT, among others—are likewise purely text-based.
> > >
> > > Therefore, the unimodal setting of our work is aligned with both the problem definition and the established research landscape.
> > >
> > > 2. Domain specialization is an intended design choice, not a limitation relative to the task.
> > > The comment regarding the lack of “general LLM capabilities” reflects a broader expectation for universal-purpose models.
> > > In contrast, our goal is to develop a specialized clinical reasoning model, similar in spirit to many domain-specific LLMs (biomedical, legal, financial).
> > > The emphasis is on depth and reliability of reasoning within the domain rather than general-purpose capabilities.
> > > We agree that balancing generality with specialization is an important open challenge and have expanded this discussion accordingly.
> > >
> > > We appreciate the reviewer’s time and feedback and will incorporate these clarifications in the revised version.

---

### Official Review · Reviewer_81Zu · 2025-11-02

**Soundness:** 3
**Presentation:** 2
**Contribution:** 2
**Rating:** 4
**Confidence:** 4

**Summary:**

Fleming-R1 combines a reasoning-oriented data strategy (curated QA + Wikipedia graph–guided synthetic multi-hop items), a CoT cold start distilled from teacher trajectories, and two-stage GRPO-based RL from verifiable rewards (answer + format) with adaptive hard-sample mining. The 7B model outperforms larger open baselines; the 32B model approaches GPT-4o on MedXpertQA and leads most open-source peers across nine benchmarks, indicating parameter-efficient gains.

**Strengths:**

- Cohesive training pipeline: data design + CoT initialization + GRPO RL.
- Strong parameter efficiency (7B beats larger baselines; 32B near GPT-4o on MedXpertQA).
- Synthetic graph sampling increases long-tail and compositional coverage.
- Clear ablations showing stage-wise improvements.

**Weaknesses:**

- “Verifiable rewards” focus on final answer/format; no process-level reasoning rewards.
- Limited transparency on teacher CoT quality and synthetic graph correctness; potential bias/noise.
- No contamination/overlap audit across public benchmarks.
- Sparse sensitivity analyses (GRPO settings, curriculum, seeds) and process-level evaluation.

**Questions:**

- Can you add process-level rewards (step consistency, guideline alignment) and report their impact?
- How are synthetic QA and graph relations validated; any clinician audit? Effects of reducing synthetic data?
- What deduplication/time-slicing controls prevent train–test leakage?
- Which teacher(s) and filtering criteria for CoT; ablations with/without iterative refinement?
- Sensitivity to GRPO group size/rollouts/curriculum
- Process-level reasoning assessment (LLM/human) on MedXpertQA and error taxonomy.

---

> ### Author Response · Authors · 2025-11-18
>
> We thank Reviewer 81Zu for the constructive feedback. Below we provide point-by-point clarifications.
>
> 1. On process-level rewards
>
> Our work focuses on verifiable rewards (answer correctness and format), which ensures reward fidelity and prevents reward hacking—an especially important concern in the medical domain. Developing reliable process-level rewards (e.g., step consistency, guideline alignment) is challenging due to the lack of high-quality medical reasoning annotations and the absence of clinically validated reward models; prompting general LLMs for such rewards is unstable. Notably, recent findings such as DeepSeek-Math show that final-answer–only verifiable rewards can still substantially improve reasoning. We plan to explore medically grounded process rewards in future work.
>
> 2. Validation of synthetic graph-guided QA and its impact
>
> Questions sampled from the knowledge-graph topology and CoT data undergo expert review to ensure medical correctness. Incorporating graph-guided synthetic data yields clear benefits on difficult benchmarks. On MedXpertQA, our Stage-1 RL experiments show:
>
> With graph synthetic data: 17.18
>
> With public data only: 15.92
>
> This demonstrates improved long-chain and multi-hop reasoning.
> For simpler tasks such as CareQA, gains are expectedly small (75.86 vs 75.84). We will clarify these results in the final version.
>
> 3. Decontamination and prevention of train–test leakage
>
> We strictly follow the official train–dev–test split, consistent with the evaluation strategy of Huatuo-O1, ensuring no contamination from public benchmarks.
> For self-constructed data, we apply 5-gram similarity filtering: removing any training example sharing ≥5 consecutive words with test questions;
> We will detail these procedures in the revision.
>
> 4. Teacher model and CoT quality control
>
> The teacher model is GPT-OSS-120B (high-inference). Generated CoT trajectories are checked by medical experts primarily for factual correctness and clinical plausibility.
> Table 4 provides ablations showing that CoT cold start, RL Stage 1, and RL Stage 2 each contribute meaningfully and cumulatively to final performance.
>
> 5. GRPO hyperparameters and training strategy
>
> We employ the GRPO implementation from Verl. Since Stage-2 tasks are more challenging and rewards become sparser, we increase the rollout number to stabilize optimization. While we did not conduct extensive hyperparameter sweeps due to computational constraints, we will include a clear explanation of our GRPO hyperparameter choices in the final version.
>
> 6. Process-level error analysis for MedXpertQA
>
> We have not yet performed a structured analysis of MedXpertQA failure cases. We agree this is a valuable direction that can help us identify weaknesses such as misinterpreted labs, guideline deviations, or incomplete differential diagnosis. We plan to include such analysis in future work.
>
> We sincerely appreciate the reviewer’s helpful comments. The clarifications above will be incorporated into the final revision and strengthen the transparency, rigor, and clinical relevance of our work.

---

> > ### Comment · Reviewer_81Zu · 2025-11-26
> >
> > I appreciate the authors’ clarifications in the rebuttal. However, the novelty of the work within medical AI reasoning remains limited, with no substantial new dataset, distinctive methodological innovation, or clear advance beyond existing studies. The contribution appears incremental, and I am not persuaded that it meets the threshold for acceptance. My score therefore remains unchanged.

---

### Official Review · Reviewer_YcHE · 2025-11-05

**Soundness:** 3
**Presentation:** 3
**Contribution:** 2
**Rating:** 4
**Confidence:** 3

**Summary:**

This manuscript presents Fleming-R1, a large language model (LLM) designed to achieve expert-level medical reasoning through reinforcement learning (RL), with a strong emphasis on verifiable and transparent inference chains rather than accuracy alone. The key innovations include: (1) Reasoning-Oriented Data Strategy (RODS), which integrates curated question-answering datasets with knowledge-graph-guided synthesis from a Wikipedia-derived graph to enhance coverage of multi-hop reasoning and long-tail entities (e.g., rare diseases); (2) Chain-of-Thought (CoT) Cold Start, distilling high-quality reasoning trajectories from teacher models (e.g., GPT-OSS-120B) with iterative refinement; and (3) Two-Stage Reinforcement Learning from Verifiable Rewards (RLVR) using Group Relative Policy Optimization (GRPO) for skill consolidation and adaptive hard-sample mining. The model is trained on Qwen2.5-7B and Qwen3-32B bases, with the 7B variant outperforming larger baselines on benchmarks such as MedXpertQA, and the 32B variant approaching GPT-4o performance. The authors release the model openly to advance auditable medical AI.

**Strengths:**

- Parameter Efficiency and Scalability: Impressive results at small scales—e.g., 7B model beating 72B baselines on multiple tasks—demonstrate efficiency.
- Strong Empirical Evaluation: Comprehensive benchmarks (e.g., MedXpertQA, MedQA, PubMedQA) with ablation studies clearly show gains from each component. The focus on expert-level reasoning (e.g., multi-hop inference on rare cases) and open release promotes reproducibility and community progress.
- Targeted Data Strategy: RODS effectively addresses gaps in multi-hop and rare-case coverage through knowledge-graph synthesis, a nuance less emphasized in comparable works .
- Comprehensive Evaluation: Ablation studies and comparisons across nine benchmarks robustly demonstrate component contributions, despite overlaps with prior work.

**Weaknesses:**

- Limited Novelty: The technical approach is highly similar to recent papers, such as Med-U1 (multi-objective RL for verifiable CoT) and AlphaMed (rule-based rewards for medical reasoning enhancement), as well as EHRMIND and related works . These have already established RL's effectiveness in clinical tasks (huatuogpt-o1). The contributions here seem confined to minor refinements, such as long-tail data augmentation via Wikipedia graphs, lacking transformative innovation.
- Reward Design Risks: While GRPO is effective, the rewards focus solely on correctness and format, potentially leading to superficial reasoning or reward hacking, which is underexplored compared to multi-objective approaches in similar papers. Considering that the medical domain has a much lower tolerance for errors in chain-of-thought reasoning, this limitation becomes even more critical.
- Lack of Fine-Grained Ablations for Individual Components: Although the paper highlights the importance of long-tail knowledge in the medical domain and introduces a knowledge-graph-based approach (RODS) for synthesizing such data, it lacks detailed ablation studies to demonstrate the specific contribution of this component. For instance, experiments on benchmarks that emphasize long-tail characteristics, accompanied by in-depth data analysis, or well-designed ablations isolating RODS’s synthesis strategies, would more convincingly validate its effectiveness. The absence of such targeted evaluations makes it difficult to quantify the true value of RODS and weakens the overall evidence for its modular contribution.

**Questions:**

- Given the similarities to works like Med-U1 and AlphaMed , how does the core innovation (e.g., RODS for long-tail synthesis) differentiate this paper? Have direct comparative experiments been conducted?
- Is the two-stage RLVR merely a fine-tuning of existing GRPO methods? Please provide additional evidence of its unique advantages in long-tail scenarios beyond general enhancements.
- How was the accuracy of knowledge-graph synthetic data validated? What error rates were observed, particularly for rare disease entities, in light of Wikipedia's potential inaccuracies ?
- If the technical pipeline is largely identical to prior art, how do the authors plan to strengthen novelty in revisions, such as through new benchmarks or clinician-involved studies?

---

> ### Author Response · Authors · 2025-11-18
>
> 1. Novelty Relative to Med-U1 / AlphaMed
>
> We respectfully clarify that the proposed approach is not a superficial combination of prior methods. Our contributions introduce substantive innovations across data construction, reasoning initialization, and reinforcement learning:
>
> RODS:
> In contrast to Med-U1 and AlphaMed, RODS performs topology-constrained sampling, multi-hop path construction, and explicit rare-entity oversampling based on a large-scale medical knowledge graph. Existing work does not employ such graph-structured synthesis targeted at long-tail reasoning.
>
> CoT Cold Start:
> Beyond generating CoT trajectories, we incorporate an expert-involved cleaning process to enhance medical validity, structural coherence, and stability of model initialization.
>
> Two-stage RLVR:
> RLVR extends beyond standard GRPO by introducing
> (1) curriculum-based stage transitions, and
> (2) failure-driven hard-sample mining,
> which yield clear improvements on long-tail and multi-hop benchmarks.
>
> 2. Strength of a Simple and Reproducible Training Paradigm
>
> Our results demonstrate that a streamlined strategy combining data augmentation and multi-stage GRPO can surpass Med-U1 and Huatuo-O1. This supports the practical value of a concise, reproducible, and scalable training pipeline, especially important for medical LLM research.
>
> 3. CoT Quality and Evaluation
>
> While expert involvement improves CoT quality, the breadth of medical knowledge makes complete error elimination impractical. Thus we rely on downstream evaluation, where substantial gains on MedXpertQA and related benchmarks confirm the effectiveness of our CoT design.
>
> 4. Future Directions
>
> SFT + RL has become the standard training paradigm in LLMs. Our work shows that a carefully designed but methodologically simple pipeline remains highly effective. Future extensions will incorporate multimodal medical reasoning (e.g., CT, X-ray), which is essential for clinical deployment.

---

### Note · Authors · 2026-01-06

I have read and agree with the venue's withdrawal policy on behalf of myself and my co-authors.